# The Nuts and Bolts of SARS-CoV-2 Spike Receptor-Binding Domain Heterologous Expression

**DOI:** 10.3390/biom11121812

**Published:** 2021-12-02

**Authors:** Mariano Maffei, Linda Celeste Montemiglio, Grazia Vitagliano, Luigi Fedele, Shaila Sellathurai, Federica Bucci, Mirco Compagnone, Valerio Chiarini, Cécile Exertier, Alessia Muzi, Giuseppe Roscilli, Beatrice Vallone, Emanuele Marra

**Affiliations:** 1Evvivax Biotech, Via di Castel Romano 100, 00128 Rome, Italy; roscilli@takisbiotech.it; 2Institute of Molecular Biology and Pathology (IBPM), National Research Council, c/o Department of Biochemical Sciences “Alessandro Rossi Fanelli”, Sapienza, University of Rome, P. le Aldo Moro, 5, 00185 Rome, Italy; lindac.montemiglio@uniroma1.it; 3Takis Biotech, Via di Castel Romano 100, 00128 Rome, Italy; vitagliano@takisbiotech.it (G.V.); luigidxx@gmail.com (L.F.); sellathurai@takisbiotech.it (S.S.); bucci@takisbiotech.it (F.B.); chiarini@takisbiotech.it (V.C.); muzi@takisbiotech.it (A.M.); 4Neomatrix Biotech, Via di Castel Romano 100, 00128 Rome, Italy; compagnone@takisbiotech.it; 5Department of Biochemical Sciences “Alessandro Rossi Fanelli”, Sapienza, University of Rome, P. le Aldo Moro, 5, 00185 Rome, Italy; cecile.exertier@uniroma1.it (C.E.); beatrice.vallone@uniroma1.it (B.V.)

**Keywords:** SARS-CoV-2, receptor-binding domain, COVID-19, spike protein, heterologous expression, protein production

## Abstract

COVID-19 is a highly infectious disease caused by a newly emerged coronavirus (SARS-CoV-2) that has rapidly progressed into a pandemic. This unprecedent emergency has stressed the significance of developing effective therapeutics to fight the current and future outbreaks. The receptor-binding domain (RBD) of the SARS-CoV-2 surface Spike protein is the main target for vaccines and represents a helpful “tool” to produce neutralizing antibodies or diagnostic kits. In this work, we provide a detailed characterization of the native RBD produced in three major model systems: *Escherichia coli*, insect and HEK-293 cells. Circular dichroism, gel filtration chromatography and thermal denaturation experiments indicated that recombinant SARS-CoV-2 RBD proteins are stable and correctly folded. In addition, their functionality and receptor-binding ability were further evaluated through ELISA, flow cytometry assays and bio-layer interferometry.

## 1. Introduction

At the end of 2019, a novel respiratory pathogen responsible for the COVID-19 disease, namely, severe acute respiratory syndrome-related coronavirus (SARS-CoV-2), emerged in Wuhan, China [1]. Only three months later, the virus spread worldwide causing one of largest outbreak of the century that rapidly progressed into a pandemic with more than 244 million of confirmed cases and 496 million deaths as of October 27th [2]. In response to this exceptional situation, an enormous effort has been made by the scientific community to study and characterize the pathogen and to quickly develop safe and effective prophylactic and therapeutic drugs.

SARS-CoV-2 is an enveloped virus whose surface is decorated with an integral membrane protein (M), an envelope protein (E), a surface spike protein (S) and an additional unexposed structural nucleocapsid protein (N) [3,4]. Among those, the Spike protein is critical for the recognition of the host-cell receptors and for mediating viral entry; therefore, it represents the most studied viral component and the best candidate for drug targeting [5,6]. The 140 kDa SARS-CoV-2 S protein is organized into two major subunits (S1 and S2) connected by a furin-cleavage site [7]. The S1 subunit contains the receptor-binding domain (RBD; aa 319–541), a 25 kDa domain that is directly involved in the interaction with the Angiotensin-converting enzyme 2 (ACE2) [8,9]. The RBD contains nine cysteines, including eight that form disulfide-bridges involved in the RBD fold. In addition, the domain displays two N-glycosylation sites (Asn_331_ and Asn_343_) known to participate in folding, stability and function [10,11,12]. Mutations occurring within this domain are constantly monitored to predict the emergence of novel variants that could be naturally selected and quickly spread, such as the recent isolated alpha (B.1.1.7), beta (B.1.351), gamma (P.1) and delta (B.1.617.2) variants of concern [13,14,15,16].

The RBD, as an isolated protein, is broadly used in different types of clinical and medical applications (serological tests, vaccine formulation, etc.) [17,18,19,20]; therefore, its in vitro production is of paramount importance. Mammalian and insect cells are the model systems of election used for the heterologous expression of SARS-CoV-2 RBD due to its intrinsic structural complexity. Attempts have also been made using other systems, such as *Pichia pastoris* [21] or *Nicotiana benthamiana* [22]. Although *Escherichia coli* (*E. coli*) represents the most common organism employed for the expression of recombinant proteins, its usage is not recommended for challenging targets that require complex folding and/or post-translational modifications such as the RBD. Nevertheless, *E. coli* gathers many technical and practical advantages (e.g., low costs and easy handling) compared to other model systems that could be beneficial both for research scale and large industrial production [23].

In this study, we present a structural and functional comparison of the native RBD of SARS-CoV-2, recombinantly produced in the three major and most frequently used expression systems (*E.* insect and mammalian HEK-293 cells), analyzing advantages and drawbacks of each preparation. The characterization of recombinant RBD proteins is of the utmost relevance in drug design to tackle the COVID-19 pandemic.

## 2. Materials and Methods

### 2.1. RBD Protein Production in E. coli

The SARS-CoV-2 Spike Receptor Binding Domain sequence (aa 319–541, Uniprot ID P0DTC2) was cloned with a C-terminal 6×-His tag into a pET-21a(+) plasmid. *E. coli* BL21 Star^TM^ (DE3) (genotype: F^-^*omp*T *hsd*S_B_ (r_B_^−^, m_B_^−^) *galdcmrne*131) competent cells (purchased from ThermoFisher Scientific, Catalog # C601003, Waltham, MA, USA) and *E. coli* Lemo21 (DE3) (genotype: *fhuA2 [lon] ompT gal (λ DE3) [dcm] ∆hsdS/pLemo*(Cam^R^)) competent cells (purchased from New England Biolabs Catalog # C2528J, Ipswich, MA, USA) were transformed with 100 ng of the plasmid of interest. A single colony was incubated in 15 mL of starter culture (LB) with Ampicillin (Mannheim, Germany) (BL21 Star strain) or Ampicillin/Chloramphenicol (Burlington, MA, USA) (Lemo21 strain), grown at 37 °C on agitation overnight. The starter culture was successively inoculated in 500 mL (LB) with antibiotics incubated at 37 °C until mid-log phase (OD_600_ = 0.6 to 0.8). Protein expression was induced with 0.5 mM isopropyl β-D-1-thiogalactopyranoside (IPTG) at 30 °C or 37 °C for 4 h on agitation. Cells were harvested at 6000 rpm for 10 min, washed once with 50 mM Tris·HCl (pH 8.0) and then further centrifuged. The pellet was resuspended in a solution containing 50 mM Tris·HCl and 500 mM NaCl (pH 8.0), also containing a protease inhibitor cocktail (Roche 11836170001, Mannheim, Germany), prior to sonication. The suspension was then centrifuged at 11′000 rpm for 45 min to separate soluble and insoluble fractions. The pellet containing the RBD target protein was resolubilized in an extraction buffer containing 50 mM Tris·HCl, 500 mM NaCl, 20% glycerol, 10 mM β-mercaptoethanol and 8 M urea (pH 8.0). The washed inclusion bodies were shortly sonicated and left for 1 h at room temperature (RT) or O/N at 4 °C on agitation. The protein was purified using IMAC (His-Trap, Cytiva, Marlborough, MA, USA) under denaturing conditions (Elution buffer: 50 mM Tris·HCl, 500 mM NaCl, 20% Glycerol, 10 mM β-Mercaptoethanol, 6 M urea and 300 mM imidazole; pH 8.0). Eluted fractions were analyzed by SDS-PAGE and firstly dialyzed overnight against buffer containing 50 mM Tris·HCl, 500 mM NaCl, 20% Glycerol, GSH-GSSG (3 mM: 1 mM) and 2 M urea (pH 8.0) with slow agitation. The day after, the protein solution was dialyzed against 50 mM Tris·HCl, 500 mM NaCl and 1 mM TCEP (pH 8.0) or PBS 1 × (pH 7.4) for 4 h. The purified *E. coli*-RBD sample protein was quantified by UV-visible spectroscopy, aliquoted and stored at −80 °C.

### 2.2. RBD Protein Production in Insect Cells

The SARS-CoV-2 Spike Receptor Binding Domain sequence (aa 319–541, Uniprot ID P0DTC2) was cloned into a pFAST-bac1 plasmid downstream of the gp64 signal sequence to promote secretion, along with a C-terminal 8×-His tag for affinity purification. A total of 100 ng of plasmid was transformed into DH10Bac competent cells (MAX Efficiency™ DH10Bac Competent Cells, Gibco #10361012, Waltham, MA, USA) for bacmid DNA production. Each bacmid, extracted from 3 mL of an O/N colony culture, was diluted in a final volume of 220 μL of Sf900 III medium and then combined with a mix of 10 μL of XtremeGene (Cellfectin™ II Reagent, Gibco #10362100, Waltham, MA, USA) in 100 μL of Sf900 III medium. This solution was left for 15 min at room temperature to allow complex formation, according to the manufacturer’s protocol. For transfection, the latter solution was added dropwise onto Sf21 cells (purchased from Gibco #11497013) previously plated on a 6-well plate at 1.0 × 10^6^ cells/well confluency. At 60 h post-transfection, the supernatant containing the first generation of recombinant baculovirus (V_0_) was harvested and amplified to obtain a high titer of the virus. Hi-5 cells (BTI-TN-5B1-4) (purchased from Gibco #B85502) were cultured in Express Five™ SFM (Serum-Free Media) medium (Gibco # B85502 Expression Systems) at a cell density of 0.5 × 10^6^ cells/mL and infected with recombinant virus. The cells were kept at 27 °C and 130 rpm for protein expression. After 72 h, the supernatant containing the secreted RBD was collected and subjected to IMAC (His-Trap Excel, Cytiva, Marlborough, MA, USA). The RBD was eluted using 50 mM Tris·HCl, 150 mM NaCl and 300 mM imidazole (pH 8.0). Eluted fractions were analyzed on 4–12% SDS-PAGE and dialyzed overnight against 50 mM Tris·HCl and 150 mM NaCl (pH 8.0) with slow agitation. The purified RBD protein was quantified by UV-visible spectroscopy, aliquoted and stored at −80 °C.

### 2.3. RBD Protein Production in HEK-293 Cells

The C-terminal 6×-His tagged SARS-CoV-2 RBD fragment (aa 319–541, Uniprot ID P0DTC2) was cloned downstream of the Ig Kappa chain-signal peptide for expression as secreted protein in mammalian cells (Expi293, purchased from ThermoFisher Scientific Catalog #A39241). Cells were transfected at a concentration of~3 × 10^6^/mL with 1 µg of DNA per milliliter of cell culture. Feed enhancers and PEN-STREP were added to the cells after 20 h and 24 h, respectively. Cells were left in agitation at 37 °C for 1 week before clarification by centrifugation at 12,700 rpm After filtration, the RBD-containing supernatant was purified by affinity chromatography on a 1 mL INDIGO column (Cube Biotech, Monheim, Germany). The sample was diluted with binding buffer (20 mM NaPi (pH 7.4) and 500 mM NaCl) and loaded at a 1 mL/min flowrate. Elution was carried out in the same conditions, with a single step of 250 mM imidazole (20 mM NaPi (pH 7.4), 500 mM NaCl and 250 mM Imidazole). The eluted protein was readily dialyzed against DPBS 1×. The isolated RBD protein was quantified by UV-visible spectroscopy (Eppendorf BioSpectrometer® fluorescence, 230 V/50–60 Hz #6137000006. Eppendorf SE, Hamburg, Germany), aliquoted and stored at −80 °C.

### 2.4. Mass-Spectrometry Analysis

SARS-CoV-2 RBD recombinant protein(s) molecular weight and primary amino acid sequence were determined by MALDI-MS and by peptide mass fingerprint (PMF), respectively. The determination of the molecular weight was achieved by a MALDI mass spectrometry analysis on a MALDI Ultraextreme (Bruker, GmbH, Billerica, MA, USA) in positive linear mode. A volume of 30 μL of the sample was desalted by diafiltration using Amicon filters with 3.5 kDa MWCO or by Zip Tip C18 (Millipore, Burlington, MA, USA) and 2 μL of the sample was mixed with a solution of the matrix superDHB. A volume of 2 μL of the resulting solution was deposited on the target plate and left to dry in the air.

In order to acquire information on the primary amino acid sequence, an aliquot of each sample was reduced, alkylated, digested with trypsin and analyzed by RP-UHPLC-MS/MS. An RP-UHPLC-MS analysis was performed on a Q-Exactive HF-X (ThermoFisher Scientific) mass spectrometer coupled with an UHPLC Ultimate 3000 RSLCnano System (ThermoFisher Scientific). A volume of 1 μL of the resulting peptide mixtures was injected on a column EasySpray PepMap RSLC C18 100 Å 2 μm, 75 μm × 15 cm (Thermo Fisher Scientific). The column oven was maintained at 35 °C; the analysis was carried using a gradient elution (phase A, 0.1% formic acid in water; phase B, 0.1% formic acid in acetonitrile). The flow rate was maintained at 300 nL/min. The mass spectra were acquired using a “data dependent scan”, able to acquire both the full mass spectra in high resolution and to “isolate and fragment” the twelve ions with the highest intensity present in the mass spectrum. The raw data were analyzed using Biopharma Finder 2.1 software from ThermoFisher Scientific Waltham, MA, USA.

### 2.5. SDS-PAGE and Western Blot

The purified proteins (500 ng) were analyzed on 4–12% NuPAGE Bis-Tris gels (Life Technologies, Carlsbad, CA, USA) under reducing conditions, followed by Coomassie Brilliant Blue staining (Invitrogen LC6060, Waltham, MA, USA). For the Western blot analysis, gels were electroblotted onto nitrocellulose membranes (Bio Rad). The blots were incubated with primary antibodies in 5% non-fat dry milk in PBS plus 0.1% Tween20 overnight at 4 °C. Detection was achieved using horseradish peroxidase-conjugate secondary antibody anti-rabbit and anti-mouse (Bio Rad #1706516, #1706515, Hercules, CA, USA) and visualized with ECL (Cytiva RPN2232, Amersham Place, England). The images were acquired by using a ChemiDoc™ Touch Imaging System (Bio Rad, Hercules, CA, USA) and analyzed by Image Lab software (Bio Rad).

### 2.6. Antibodies

The primary antibodies used in this study were: rabbit anti-SARS-CoV-2 Spike S1 Subunit (Sino Biological, 40150-T62, Beijing, China) and mouse anti-His Tag (Invitrogen MA1-21315). Secondary antibody used were: Horseradish peroxidase-conjugate anti-rabbit and anti-mouse (Bio Rad #1706516, #1706515).

### 2.7. Densitometric Analysis

The intensities of bands corresponding to the RBD proteins were measured using Gel Doc 2000 and Image Lab software (Bio-Rad, Hercules, CA, USA) in order to measure the protein expression levels. Briefly, the blots were acquired using the Gel Doc 2000 apparatus; the images were imported into Image Lab software (Bio-Rad, Hercules, CA, USA); the contrast was adjusted such that the bands were clearly visible on the blot image; the area around each band was selected; the background intensity was subtracted from the blot image; the bands were then selected by drawing a tight boundary around them; the intensities of the selected bands were exported in an excel file format, which was used to perform further analyses.

### 2.8. ELISA Assay

ELISA plates were coated with different concentrations of *E. coli*-RBD, Insect-RBD and HEK-293-RBD proteins. After washing and blockading of the free protein-binding sites with PBS—0.05% Tween20—and 3% BSA, different concentrations of rat serum (immunized with COVID-*e*Vax vaccine) or anti-SARS-CoV-2 Spike S1 Subunit antibody (Sino Biological, 40150-T62) were added to each well and incubated overnight at 4 °C in PBS—0.05% Tween20—and 1% BSA. After washing, AP-conjugated goat anti-rat IgG antibody (SIGMA A8438) or AP-conjugated goat anti-rabbit IgG antibody (SIGMA A8025, Burlington, MA, USA) was added and the plates were further incubated for 1 h at RT. Finally, 3,3′,5,5′-Tetramethylbenzidine (TMB) Liquid Substrate System (Sigma T8665) or alkaline Phosphatase Yellow (pNPP) Liquid Substrate System for Elisa (Sigma P7998) was added as a substrate. After 30 min, the TMB reaction was stopped with the stop reagent for TMB substrate (Sigma S5814) and the absorbance was measured at 450 nm, while the pNPP reaction was measured at 405 nm at different time points. The optimal cut-off value was determined using the formula Cutoff = 2× + 3 S.D., where x is the mean and S.D. is the standard deviation of three independent negative-control readings. To discriminate positive results from background readings, the obtained highest cut-off value was chosen to be represented on the graphs.

### 2.9. FACS

Vero E6 cells were incubated with the RBD protein (0.45 μg/mL, final concentration) followed by incubation with human anti-RBD antibody (primary antibody) (40150-D003, Sino Biological, Beijing, China and goat anti-human IgG AF488-conjugated antibody (secondary antibody) (A-11013, Thermo Fisher Scientific). Staining with only secondary antibody was used to determine the level of background due to non-specific antibody binding. Each staining step was performed at 4 °C for 20 min in FACS buffer. The samples were run on a CytoFlex flow cytometer (Beckman Coulter). The analyses were performed using CytExpert software (Beckman Coulter, Brea, CA, USA).

### 2.10. CD Static Spectra and Thermal Denaturation

The far-UV (200–250 nm) circular dichroism (CD) spectra of the SARS-CoV-2 HEK-293-RBD and Insect-RBD were recorded using 0.6 mg/mL of protein solution in PBS (pH 7.4) and 0.3 mg/mL of protein solution in 50 mM Tris·HCl and 150 mM NaCl, respectively. The CD spectra of *E. coli*-RBD were monitored at a 0.3 mg/mL protein concentration in 50 mM Tris·HCl, 150 mM NaCl, 1 mM TCEP and 20% glycerol (pH 8). All CD static spectra were collected at 20 °C and scanned at 50 nm/min, using a 0.1 cm path length quartz cuvette (Hellma, Plainview, NY, USA) and a JASCO-815 spectropolarimeter equipped with a Jasco programmable Peltier element (Jasco, Easton, MD, USA). For each sample, five scans were averaged and the scans corresponding to the buffer solution were averaged and subtracted from the sample spectra. The results are expressed as the molar ellipticity ([Θ]). The formula used to calculate the molar ellipticity was [θ] = (θ × MW)/(C × L × 10), where [θ] is the molar ellipticity, θ is the experimental ellipticity in mdeg, MW is the molecular weight of the protein in Daltons, C is the protein concentration in mg/mL and L is the path length of the cuvette in cm. The secondary structure composition was assessed using the BeStSel analysis server (Budapest, Hungary) [24,25].

The CD thermal denaturation experiments were followed at 222 nm, heating from 20 °C to 80 °C at a rate of 1 °C min^−1^ controlled by a Jasco programmable Peltier element (Jasco, Easton, MD, USA). The dichroic activity at 222 nm and the photomultiplier voltage (PMTV) were continuously monitored in parallel every 1.0 °C [26]. The data were fitted to a standard two-state denaturation [27], according to Equation (1).
(1)$$∆G_{D\text{-}N}=∆H_{T_{m}}\left(1-\frac{T}{T_{m}}\right)+∆C_{p}\;\left[T-T_{m}-\left(T\;ln\frac{T}{T_{m}}\right)\right]$$
where ∆*G_D-N_* is the free energy of the unfolding process, *T_m_* is the melting temperature that corresponds to midpoint of the thermal denaturation, ∆*H_Tm_* is the enthalpy of denaturation at the transition midpoint and ∆*C_p_* is the change in heat capacity of denaturation. The latter parameter is related to the amount of hydrophobic area that becomes exposed to solvent upon unfolding. In a first approximation, the thermodynamic parameters of unfolding were estimated using the ∆*C_p_* value reported for a globular protein of similar size, namely, α-chymotrypsin (241 amino acids) [28], and are presented in Appendix A. All denaturation experiments were performed in triplicate.

### 2.11. Size-Exclusion Chromatography

Analytical gel filtration chromatography was performed using a Superdex 200 Increase 10/300 GL SEC column (Cytiva, Marlborough, MA, USA) coupled to an HPLC system (Azura System, Knauer-Berlin, Germany) equipped with a UV-vis absorbance detector (Smartline 2520, Knauer-Berlin, Germany). The column was equilibrated with 50 mM Tris·HCl (pH 8.0), containing 150 mM NaCl. In total, 40 µg of HEK-293-RBD, 47 µg of Insect-RBD and 40 µg of *E. coli*-RBD were injected into the column and eluted at a flow rate of 0.75 mL/min in isocratic mode. The elution profile was followed at 280 nm at room temperature. The shape of the elution profiles and the differences among HEK-293-RBD, Insect-RBD and *E. coli*-RBD were observed reproducibly in three independent experiments.

### 2.12. Bio-Layer Interferometry (BLI)

Binding studies were carried out using the Octet Red system (Forte Bio, Fremont, CA, USA). All steps were performed at 25 °C with shaking at 600 rpm in a 96-well plate (microplate 96 well, F-bottom, black, 655209, from Greiner bio-one) containing 200 µL of solution in each well. A kinetics buffer 1× (cat. No.18-1105, Forte Bio) was used throughout this study for sample dilution and for sensor washing.

Kinetic assays were performed by first capturing ACE2-hFc using anti-human Fc Octet biosensors (Anti-human IgG Fc Capture Biosensors, cat. No. 18-5060, Forte Bio, Fremont, CA, USA). The biosensors were soaked for 10 min in 1× kinetic buffer followed by a baseline signal measurement for 60 s and then loaded with ACE2-hFc recombinant protein (10 µg/mL) for 300 s (until the biosensor was fully saturated). After a wash step in 1× kinetic buffer for 120 s, the ACE2-Fc-captured biosensor tips were then submerged for 300 s in wells containing different concentrations of antigen (RBD *E. coli* and Insect and HEK-293) to evaluate the association curves, followed by 900 s of dissociation time in kinetic buffer. The ACE2-hFc captured biosensor tips were also dipped in wells containing kinetic buffer to allow single reference subtraction to compensate for the natural dissociation of captured ACE2-hFc. Biosensor tips were used without regeneration.

The binding curve data were collected and then analyzed using data analysis software version 12.0 (FORTEBIO). The binding sensorgrams were first aligned at the last 5 s of the baseline step average. The single-reference subtraction binding sensorgrams were globally fit to a 1:1 Langmuir binding model to calculate *K_d_* values.

## 3. Results

### 3.1. Design, Expression and Purification of SARS-CoV-2 RBD in E. coli

The RBD protein (Figure 1a) was recombinantly expressed with a C-terminal 6×-His purification tag both in BL-21 Star and Lemo21 cells. The Lemo21 bacterial strain allows challenging targets, such as toxic, highly insoluble and membrane proteins, to be expressed by reducing inclusion body formation and potential inhibitory effects on cell growth, thus resulting in an increased level of properly folded products. However, only a negligible amount of the RBD was found in the soluble fraction, even exploring alternative growing conditions, including lower temperature, distinct induction times and increasing concentrations of L-Rhamnose (data not shown). The target protein was totally recovered from inclusion bodies with yields representing 5.2% (Star) and 8.1% (Lemo21) of the total protein extract (Figure 1b; Appendix A). Protein purification was carried out in the presence of denaturing agents (6 M urea) followed by a slow refolding process through an overnight dialysis against buffer containing the redox pair of oxidized and reduced glutathione to induce proper disulfide bond formation (Figure 1b; Appendix A). As shown in Figure 1c, the purified *E. coli*-RBD protein shows a high degree of purity (>90%) and migrated as a single-smeared band at the expected height on 4–12% SDS-PAGE (theoretical mass, 26052 Da). Moreover, the Western blot analysis indicated that the protein was efficiently recognized by anti-His and anti-SARS-CoV-2 Spike S1 subunit antibodies (Figure 1c). Approximately 1.25 mg of purified RBD was obtained starting from 0.5 L of bacterial culture (final yield, ~2.5 mg/L). Among distinct batches, the concentration ranged from 0.1 mg/mL (3.8 µM) to 0.3 mg/mL (11.5 µM). Concentrations higher than 0.3 mg/mL led to protein precipitation. Finally, the molecular weight and primary amino acid sequence of SARS-CoV-2 RBD purified from *E. coli* were further validated by Mass-spec analysis (Appendix A).

### 3.2. Design, Expression and Purification of SARS-CoV-2 RBD in Insect and Mammalian Cells

The RBD fragment with a C-terminal 8×-His purification tag was cloned downstream of the gp64 for expression in insect cells (Figure 1a). The generation and amplification of recombinant baculovirus were carried out in Sf21 cells, while protein expression was performed in Hi-5 infected insect cells (Figure 1b). The soluble protein of interest (POI) was secreted into a culture medium and purified through immobilized metal affinity chromatography (IMAC) using a Ni-Nta resin (Figure 1b; Appendix A). The isolated insect-RBD migrated as a single band slightly higher than 25 kDa on 4–12% SDS-PAGE (theoretical mass, 26266 Da), exhibiting a high level of purity (>95%) and it was clearly detected by immunoblotting (anti-His and anti- SARS-CoV-2 Spike S1 subunit) (Figure 1d). At a laboratory-scale, final yields were around 6.5 mg of the RBD per liter of insect cells with batch concentrations ranging from 0.25 mg/mL (9.5 µM) to 0.5 mg/mL (19 µM). The experimental mass (28936 Da) of the recombinant insect-RBD determined by the MALDI mass-spectrometry analysis (Appendix A) was higher than the theoretical one based on the amino acid composition, thus suggesting the presence of glycosylations [29].

Regarding RBD expression in mammalian cells, SARS-CoV-2 RBD flanked by a C-terminal 6×-His tag was cloned downstream the Ig Kappa chain-signal peptide responsible for protein secretion (Figure 1a). Cells were transfected with DNA and left under stirring and controlled CO_2_ atmosphere for 1 week expressing the POI. The RBD-containing medium was filtrated and the POI was purified by affinity chromatography (Figure 1b; Appendix A). The eluted protein migrated as a single, slightly diffuse band below 37 kDa, indicating that the RBD (theoretical mass, 26135 Da) contained glycosylations. Indeed, the experimental mass obtained from the MALDI-MS analysis was 31453 Da, confirming the presence of post-translation modifications, as previously reported [21,30,31] (Appendix A). Additionally, the eluted protein was efficiently detected by anti-His and anti-S1 subunit of SARS-CoV-2 Spike antibodies (Figure 1d). Around 800 mL of transfected cells yielded 58.8 mg of pure purified protein, with batch concentrations reaching up to 1.8 mg/mL (69 µM).

### 3.3. Biochemical Characterization of RBD

The RBDs produced in HEK-293, insect cells and *E. coli* were analyzed by size exclusion chromatography (SEC) (Figure 2a). *E. coli*-RBD eluted as a single and narrow peak centered at 18.7 mL, whereas the RBD proteins produced in HEK-293 and insect cells displayed elution peaks shifted to lower retention volumes owing to the presence of glycosylations. In fact, HEK-293-RBD eluted as a main peak centered at 15.3 mL, while the RBD produced in insect cells eluted as a major one centered at 16 mL and a minor one at 14.6 mL. The presence of two peaks in the insect-RBD elution profile suggests the existence of at least two populations of the protein showing alternative glycosylation patterns and differing from the one of HEK-293-RBD. The lack of glycosylation of *E. coli*-RBD shifted the retention volumes to higher values. Altogether, all the elution peaks observed are all consistent with a ~30 kDa protein.

RBD proteins were analyzed by far-UV CD spectroscopy. In this analysis, HEK-293-RBD spectra were recorded in PBS (pH 7.4), Insect-RBD in 50 mM Tris·HCl and 150 mM NaCl (pH 8) and *E. coli*-RBD in 50 mM Tris·HCl, 150 mM NaCl, 1 mM TCEP and 20% glycerol (pH 8). The choice of working in a buffer of different composition was dictated by the stability of the samples, that was particularly critical in the case of the RBD from *E. coli*. Indeed, the latter sample, in the absence of TCEP and glycerol, or in PBS buffer, tended to precipitate.

The spectral profiles of HEK-293-RBD and Insect-RBD reported in Figure 2b are by-and-large identical, both displaying a single minimum at~206 nm and a maximum at~230 nm, which are typical of native RBDs [21]. Conversely, the far-UV CD spectrum of *E. coli*-RBD differs from those of the eukaryotic counterparts, as also observed by Mycroft-West et al. [32]. However, the analysis of the secondary structure composition returned an overall similar distribution (Figure 2c). The slight difference in the α-helical content estimation, which was particularly evident for the *E. coli*-RBD sample, is attributable to the decreased signal-to-noise ratio observed below 210 nm, which is due to the presence of 150 mM sodium chloride concentration.

The conformational stability of RBDs was investigated by means of temperature denaturation experiments. We followed the variations in far-UV CD ellipticity at 222 nm upon an increase in temperature from 293 K to 350 K (Figure 2d). The change in ellipticity, monitored for each construct, followed a sigmoidal dependence upon temperature increase, suggesting that the RBDs reversibly unfolded. Differently to what expected for a typical folded-to-unfolded transition followed by far-UV CD, the ellipticity values decreased with the increase in temperature (no loss of CD signal).

The observed denaturation curves could be well fitted to a two-state transition, according to Equation (1). The resulting *T_m_* values determined for HEK-293-RBD and Insect-RBD are identical within experimental error, yielding a mean value of 323.3 ± 1.0 K and 323.8 ± 0.4 K, respectively. The estimated *T_m_* values are consistent to what previously reported for the RBD produced in eukaryotic cells in similar ionic strength conditions [21]. *E. coli*-RBD showed lower *T_m_*, equal to 319.8 ± 0.2 K. The thermodynamic parameters derived from the analysis of the unfolded curves and presented in Appendix A confirmed the reduced thermal stability of the *E. coli*-RBD sample compared to the protein produced from eukaryotic systems. It is worth mentioning that, while most of the analyzed *E. coli*-RBD samples showed a single unfolding transition, in a limited number of cases, the denaturation curve shows a biphasic behavior, indicating the existence of an initial unfolding event preceding the main one and taking place around 305 K (data not shown). We surmise that this initial phase is likely due to a minor portion of the protein that failed refolding during sample purification from inclusion bodies, probably owing to incorrect disulfide bridges formation.

### 3.4. ELISA Assays

The functionality of the RBD protein was determined through ELISA assays using plates coated with the RBD produced either in *E. coli*, insect or in HEK-293 cells. First, to test coating conditions, 50 ng/well of RBD proteins were used for coating in 50 μL of phosphate-buffered saline (PBS) or carbonate buffer. Serially diluted (1:1000, 1:10000, 1:50000) rat sera (immunized with COVID-eVax vaccine [34]) were used to detect the optical density (OD) associated with antibody-RBD interaction under distinct conditions. Significant differences were observed between plates, suggesting that PBS buffer is the most efficient buffer for coating (data not shown).

Subsequently, the plates were coated with increasing RBD protein concentrations (ranging between 1 and 5 µg/mL) and serum from rats previously immunized with COVID-eVax [34] vaccine was applied to each plate for RBD protein binding. Of note, independently from protein concentration (1, 3 and 5 μg/mL) both insect-RBD and HEK-293-RBD were efficiently recognized by rat IgG, whereas the rat IgG–*E. coli*-RBD interaction was much lower (Figure 3a). The observed differences between the RBD produced in *E. coli* and in the insect or mammalian counterparts are probably due to the major affinity of the latter with the IgG produced in rats.

As a positive control, the interaction between the recombinant RBD and a commercial antibody against the S1 subunit of SARS-CoV-2 Spike was monitored. As shown in Figure 3b, the observed OD signal of insect-RBD was not markedly different from that of HEK-293-RBD, although, at concentrations <1 μg/mL, the RBD from insect cells showed a slightly higher binding ability compared to HEK-293-RBD (Appendix A). A positive signal was also observed for *E. coli*-RBD (3 and 5 μg/mL), although the latter showed lower binding than HEK-293 and insect RBDs. We hypothesized that this lower binding ability of RBD produced in *E. coli* may, again, be due to the presence of a sub-population of the protein that failed refolding.

### 3.5. Flow Cytometry Assay

The receptor-binding ability and functionality of RBDs produced in the three presented model systems were further investigated through flow cytometry. Vero E6 cells have been shown to express the ACE2 receptor on their apical membrane and to be susceptible to SARS-CoV-2 infection [35,36]. Thus, we tested RBD–ACE2 binding by incubating recombinant RBD proteins with cultured Vero E6 cells. Figure 4 shows that all the three studied RBDs were able to efficiently bind Vero E6 cells, while no signal was observed when cells were incubated only with antibodies (Appendix A). This result suggests that recombinant RBD proteins are efficient in recognizing ACE2.

### 3.6. Bio-Layer Interferometry Binding Assay

Finally, the binding affinity to ACE-2 receptor of the RBD produced in *E. coli*, insect and HEK-293 cells was evaluated using bio-layer interferometry (BLI). The ACE2-hFc fusion protein was immobilized onto anti-human Fc biosensor and different concentrations of RBD proteins (range, 150 nM–9.8 nM) were tested to obtain association curves. After fitting, the dissociation constant (K_d_) of ACE2-hFc to insect-RBD and to HEK-293-RBD was determined to be 7.49 × 10^−9^ M and 5.34 × 10^−10^ M, respectively, while much lower binding affinity was observed for *E. coli*-RBD (K_d_ = 1.21 × 10^−6^ M) (Figure 5a,b; Appendix A).

## 4. Discussion

The emergence of the novel SARS-CoV-2 pathogen at the end of 2019, which has quickly degenerated into a pandemic, has underlined the importance of immediate and responsive actions from the local governments, health authorities and the world scientific community in order to tackle this situation that probably represents the biggest challenge that modern society has faced. As a result, over the last two years, several vaccines have been developed and many drugs are currently under screening or evaluation in clinical trials [37,38,39]. Most of those therapeutics target the Spike protein and, more specifically, its receptor-binding domain, which is exposed on the viral envelope and that is directly involved in receptor binding and cell entry. Moreover, both full-length Spike and RBD are widely used as viral antigens for diagnostic tests, representing a critical tool for a fast response to the pandemic.

In this study, we provide technical insights into the heterologous expression, purification and characterization of the native SARS-CoV-2 RBD produced in the *E. coli*, insect and HEK-293 model systems. Bacterial RBD production was achieved by recovering the protein from the insoluble fraction and through a careful process of refolding. The efforts to increase its solubility by using the Lemo21 *E. coli* strain failed in agreement with previous attempts to produce this protein, or its ancestor (SARS-CoV RBD), in *E. coli* in its native soluble form [21,40]. After refolding, isolated *E. coli*-RBD showed a good degree of purity and was efficiently recognized by commercial antibodies. Considering the challenging target, the final obtained yields were not high, but enough to carry out most of the lab-scale downstream applications. In contrast, the production in insect and HEK-293 cells resulted in more soluble, highly glycosylated RBD proteins, with yields up to 60 mg/L. The presence of post-translational modifications (glycosylations) in the latter samples was indirectly observed by SDS-PAGE, mass-spectrometry analysis and size-exclusion chromatography. The nature and the type of glycosylation was not further investigated, but it does not affect the overall thermal stability (Appendix A). Consistently with the absence of glycosylations, we found that *E. coli*-RBD presents lower thermal stability compared to RBDs produced in eukaryotic organisms. We cannot exclude a contribution to the observed lower stability of a limited fraction of *E. coli*-RBD that failed refolding, lacking disulfide bridges. Of note, although the far-UV-CD spectrum profile of the RBD from *E. coli* appeared different from the one observed for HEK-293-RBD and insect, the overall distribution of the secondary structure composition is similar. The underestimation of the α-helical content, more pronounced in the case of *E. coli*-RBD, was caused by the lower CD signal observed below 210 nm, which is due to the partial absorption of chloride ions present in solution. Another related issue to comment on is the use of different buffer conditions for the differently produced RBD samples in the CD analysis. The choice of the buffer composition was dictated by the stability of the proteins. Even though these conditions are not optimal for a CD study, they allowed us to assess that the refolded protocol we devised to obtain the SARS-CoV-2 RBD from *E. coli* worked and that the produced RBDs were folded in those conditions, as indicated (i) by the similarity of the overall far-UV-CD shape with what previously reported for HEK, insect and *E. coli* RBDs [21,32,41], (ii) by the overall consistency in the distribution of the secondary structure composition of the proteins and (iii) by the agreement of the calculated melting temperatures with those reported in the literature in equivalent ionic strength conditions [21]. Notably, we can exclude an effect of the presence of TCEP on the differences observed between the CD spectra of *E. coli*-RBD and the ones of HEK-293- and Insect-RBDs, since the spectrum profile of the RBD from *E. coli* reported in Mycroft-West et al. [32] was obtained in the absence of the reducing agent. Moreover, exchanging the buffer conditions of HEK-RBD with those used for Insect-RBD did not alter the static CD nor the thermal unfolding profile of the sample, returning thermodynamic parameters within the experimental error (Appendix A). All considered, in our experiments, different buffer conditions may have had an effect on the signal, but they barely impacted the profiles of the static CD and of the unfolding transition.

RBD functionality was demonstrated in vitro using an enzyme-linked immunosorbent assay (ELISA). The RBD produced in *E. coli* displayed a weak binding affinity to IgG produced in rats and to commercially available antibodies, while an efficient response was observed for both insect- and mammalian-derived RBDs. Remarkably, at lower concentrations (<1 μg/mL), insect-RBD gave a slightly better signal than HEK-293-RBD. We also investigated the capability of isolated RBDs to bind the ACE2 receptor. All the RBDs produced in this work efficiently bind to Vero E6 cells, as confirmed by the FACS assay. ACE2–RBD binding was further confirmed and quantified by bio-layer interferometry, with the bacterial-RBD again displaying the lowest binding efficiency. Our data suggest that the absence of glycosylation could partially affect ACE-2 binding in vitro, as also previously observed [10]. In addition to this, we must consider that the presence of a sub-population of protein that failed refolding, as indicated by circular dichroism and ELISA assays, might also contribute to the observed lower binding efficiency of the bacterial RBD.

To summarize, this work offers a technical and practical overview of RBD production using the three most widely used expression systems, highlighting the main advantages and drawbacks, reported in Table 1. The RBD obtained from both eukaryotic systems resulted in a high-quality final bioproduct potentially eligible for diverse downstream applications (vaccine design, diagnostic kits, drug screening, etc.). However, the high costs, the time-consuming production, the requirement of specific equipment and the access to dedicated facilities could be a limitation for many laboratories or for industrial production. By contrast, the bacterial-derived RBD offers low production costs, a broader availability and easy handling as main advantages, which make it more accessible. However, limitations in the quality of the produced sample include the absence of glycosylation, that partially affects protein stability and efficiency; the presence of heterogeneous folded populations; and the relative low production yields, which may result in a final product that is not eligible for some clinical and medical applications. Overall, all the recombinantly produced RBDs represent valuable tools for research purposes against the pandemic. Recently, the expression and purification strategies described in this article have been also proved to be successful in the production of mutants of the RBD corresponding to the variants of concern.

## Figures and Tables

**Figure 1 biomolecules-11-01812-f001:**
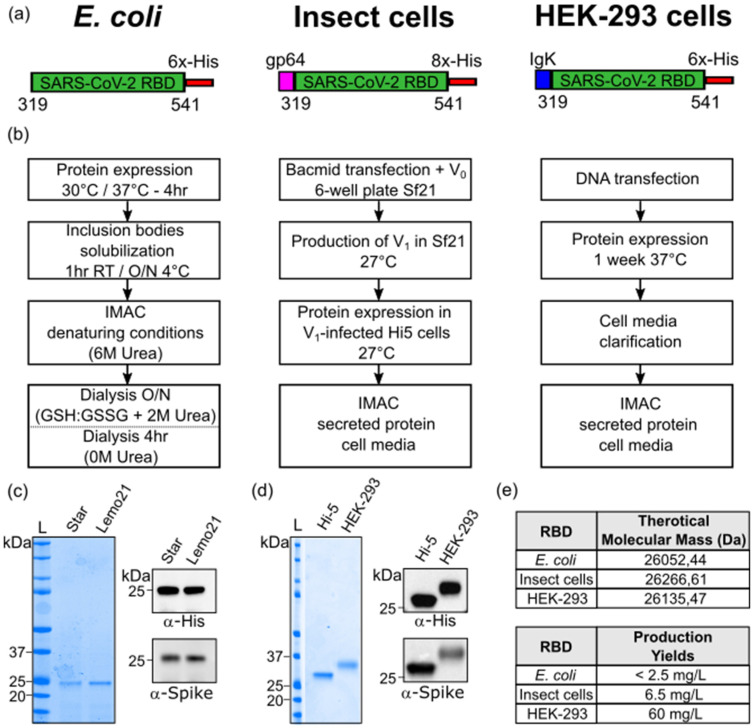
SARS-CoV-2 RBD production in *E. coli*, insect and mammalian cells. (**a**) Schematic representation of the RBD protein constructs expressed in *E. coli* (left), insect cells (middle) and mammalian HEK-293 cells (right). (**b**) Diagram summarizing the RBD recombinant expression from *E. coli* (left), insect cells (middle) and mammalian HEK-293 cells (right) and the subsequent purification. (**c**) SDS-PAGE (left panel) and Western blot analysis (right panels) of *E. coli*-purified RBD protein. (**d**) SDS-PAGE (left panel) and Western blot analysis (right panels) of RBD fragment produced in Hi-5 insect cells and mammalian HEK-293. L = molecular weight ladder. (**e**) Theoretical molecular masses calculated according to RBD amino acid composition (above) and RBD production yields (below).

**Figure 2 biomolecules-11-01812-f002:**
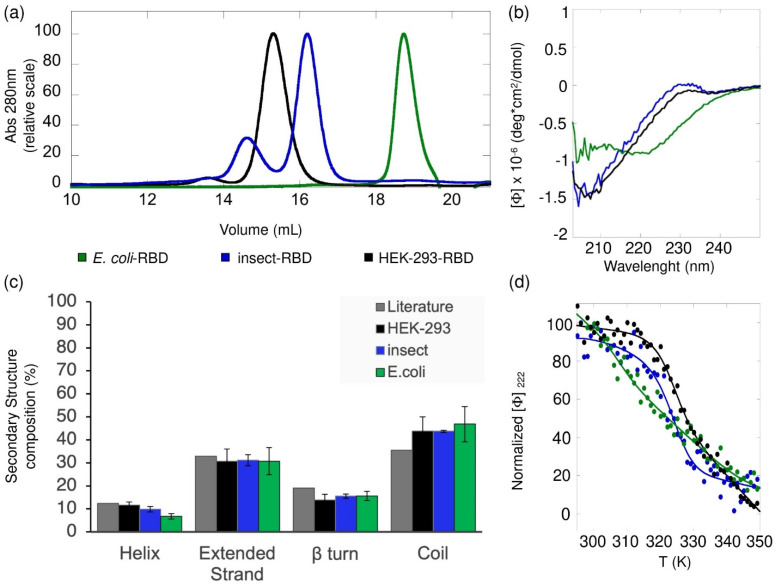
Biochemical characterization of recombinant RBD. (**a**) Gel filtration chromatographic profiles. Protein separation was performed at room temperature using a Superdex 200 Increase 10/300 GL with 40 µg of RBD produced in HEK-293 (black), 47 µg of RBD produced in insect (blue) and 40 µg of RBD produced in *E. coli* (green), each in 50 mM Tris·HCl and 150 mM NaCl (pH 8.3). (**b**) Far-UV CD spectra of RBD produced in HEK-293 (black), Insect (blue) and *E. coli* (green) cells. All spectra were collected at 20 °C, using a 0.1 cm path length quartz cuvette. (**c**) The histogram reports the distribution of the secondary structure content determined for the RBD proteins (at least three independent CD experiments (means ± standard deviation)), in comparison with the secondary structure composition of RBD reported by Lan et al. (dark grey bars, Literature) [33]. (**d**) Thermal denaturation profiles of RBD *E. coli* (green), Insect (blue) and HEK-293 (black), continuously monitored by far-UV CD at 222 nm over the range 293–350 K. Data were fitted using a two-state model. The estimated thermodynamic parameters derived from these analyses are presented in Appendix A.

**Figure 3 biomolecules-11-01812-f003:**
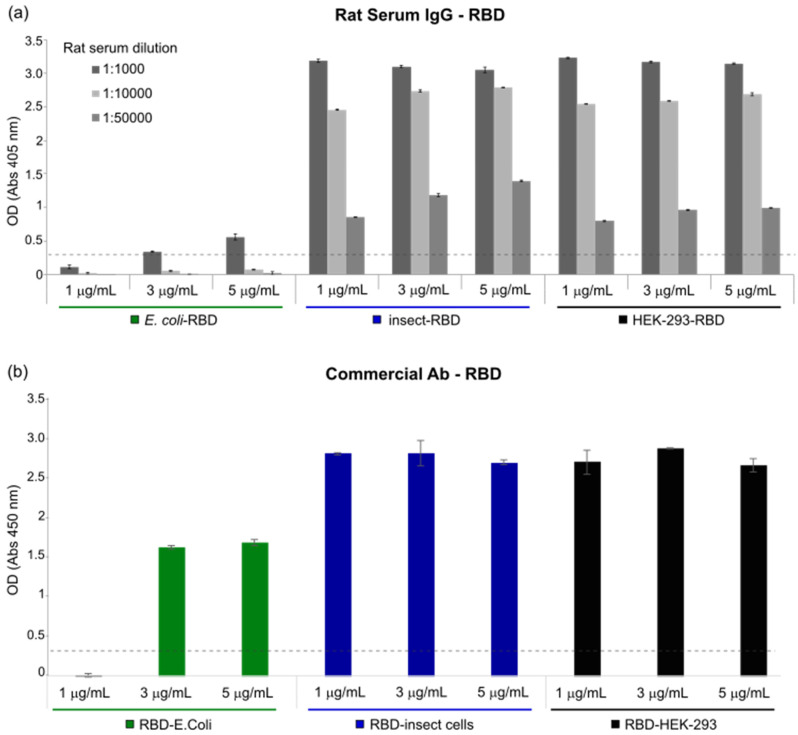
ELISA assays. (**a**) Serum from immunized rat with COVID-*e*Vax was used to compare different concentrations (1, 3 and 5 μg/mL) of the RBD expressed in *E. coli* (green), insect (blue) and HEK-293 cells (black). The *y*-axis represents the optical density (OD) measured at 405 nm, while the *x*-axis accounts for RBD concentrations and serum dilution factors (1:1000, 1:10,000 and 1:50,000). Bars indicate standard deviations. Dashed line = cut-off value. (**b**) Commercial antibody against the S1 subunit of SARS-CoV-2 Spike was used to compare different concentrations (1, 3 and 5 μg/mL) of RBD produced in *E. coli* (green), insect (blue) and HEK-293 cells (black). Optical density (OD) was measured at 450 nm and bars indicate standard deviations. Dashed line = cut-off value.

**Figure 4 biomolecules-11-01812-f004:**
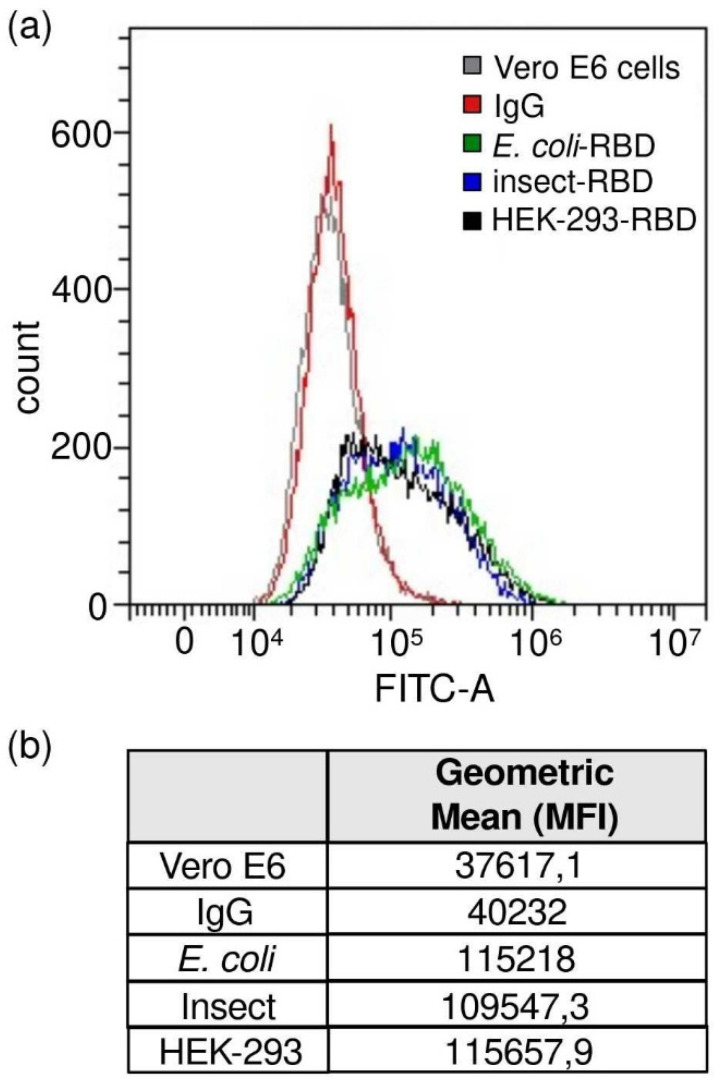
Flow cytometry assays. RBD-Vero E6 cells binding experiment. (**a**) Gray curve, Vero E6 cells alone; red curve, Vero E6 cells incubated with only secondary antibody; green curve, Vero E6 cells incubated with *E. coli*-RBD; blue curve, Vero E6 cells incubated with Insect-RBD; black curve, Vero E6 cells incubated with HEK-293-RBD. Incubation with RBD was followed by anti-RBD primary antibody and secondary antibody. (**b**) Intensity of the staining measured as geometric mean (median fluorescence intensity, MFI) value.

**Figure 5 biomolecules-11-01812-f005:**
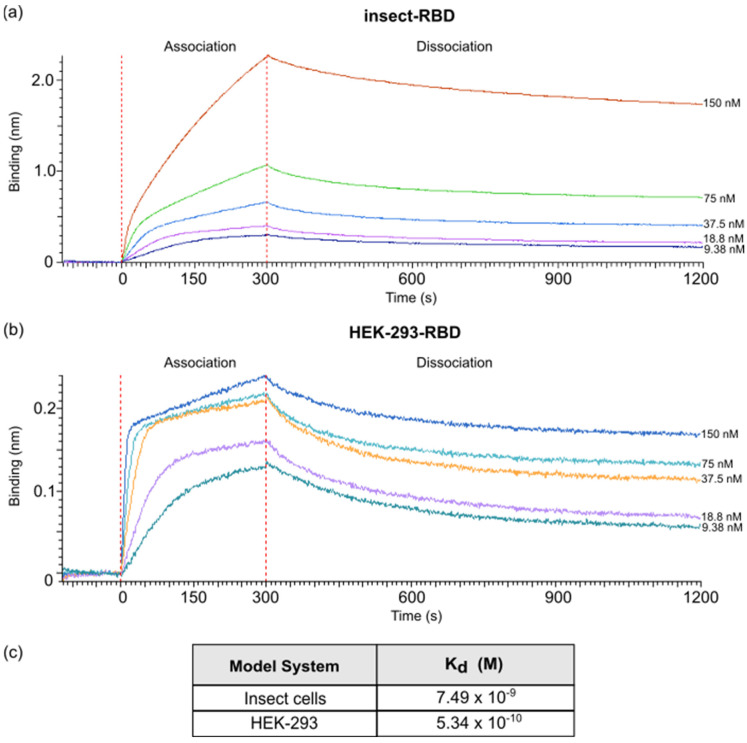
BLI measurements. (**a**) BLI profiles accounting for the binding of insect-RBD and (**b**) HEK-293-RBD to ACE2-hFc. After a baseline, the sensorgram starts with the association (0–300 s) of the RBD to the ACE2-loaded sensor, followed by the dissociation phase (900 s). (**c**) K_d_ measured values.

**Table 1 biomolecules-11-01812-t001:** Overview of the main aspects of the RBD produced in the three major model systems. * The cost per liter accounts for the resources (cell media, transfection reagents, disposables, etc.) used in the upstream and downstream processes. It does not include costs related to the cloning process, the manpower, nor the equipment usage. ** Flow cytometry assay using Vero E6-cells to measure RBD–ACE-2 binding. + (low) ++ (medium) +++ (high) ++++ (very-high).

Model System	Average Cost Per Liter *	Production Time	Yields	Soluble Expression	Folding	Glycosylation	ELISA	ACE-2 Binding (FACS) **	ACE-2 Binding (BLI)
*E. coli*	~EUR 256	≤1 week	+	NO (need for refolding)	++	NO	+	+++	+
Insect cells	~EUR 402	2–3 weeks	+++	YES	++++	YES	++++	++++	++++
HEK-293	~EUR 1325	2–3 weeks	++++	YES	++++	YES	++++	++++	++++

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
