# Peer review of "The Nuts and Bolts of SARS-CoV-2 Spike Receptor-Binding Domain Heterologous Expression"

_biomolecules, 2021, doi:10.3390/biom11121812_

Round 1

Reviewer 1 Report

The authors present a comparison of expression systems to overproduce the SARS-CoV-2 Spike Receptor Binding Domain. The manuscript is well written with biochemical, biophysical and functional data presented in logical progression. The report can inform ongoing efforts to produce new vaccines against COVID-19 as well as to develop diagnosis kits. There are issues about the consistency of the data presented. These are described below:

-Figure 2 Panel (c) shows that the lowest alpha-helix content, as determined by Far-UV CD, was the protein domain expressed in E. coli. However, panel (b) of the same figure indicates the opposite: a characteristic inflection at 222nm and a higher alpha-helix content. More worrying is the data shown in Panel (d), where the molar ellipticity of the protein produced in E. coli at 290K is approximately 2/3 of the value shown in Panel (b). 

-I wonder why the authors used different buffer for circular dichroism? On top of that, PBS and Tris-NaCl are not appropriate for this type of study. Valuable information was lost in the far-UV region (180-200nm range) due to the choice of buffer for their CD scans. They should have used a buffer lacking chloride ions to ensure a good signal/noise ratio beyond 205nm and prepared all the samples in the same buffer for all the CD scans. In addition to the fact the protein produced in bacteria was refolded and lack glycosylation, the difference in buffer composition and ion strength may also contribute to the observed differences in CD profiles the authors mentioned in the discussion section (lines 463-465). A comment about the rational for the choice of buffer for far-UV circular dichroism and an expanded discussion about the impact of the use of buffers of different composition for different samples on CD spectra profiles seems appropriate.   

- The fact CD data could not be recorded below 205nm may explain the discrepancy of secondary structure content estimated from CD data covering the 205-240 nm range compared to the crystal structure of the SARS-CoV-2 Spike Receptor Binding Domain (PDB ID 6MOJ). 

Some minor changes the authors should address are listed as follows:

  • Change "primary aminoacidic sequence" to primary amino acid sequence as the latter is the term generally accepted to refer to a protein primary structure.
  • E. coli should appear in italics (line 292).
  • Figure 2 (d) the scale of the graph does not match the range indicated in the figure legend.

Reviewer 2 Report

The manuscript “The nuts and bolts of SARS-CoV-2 Spike Receptor Binding Domain heterologous expression” by Maffei et al. reports the biochemical/biophysical comparison between Spike RBD from SARS-CoV-2 obtained from three different expression systems. Spike RBD, a key element for virus internalization, is an important biotechnological and pharmacological protein: it has been used for vaccine and diagnostic kit development. Obtaining purified protein under optimal conditions (yield, purity, functionality…) is paramount for successful downstream applications. The authors compared Spike RBD expressed using E. coli, insect cells and HEK-293 cells. They found that, although E. coli is a much cheaper and fast expression system, eukaryotic cells provide higher yield, proper folding and glycosylation pattern, and optimal interaction capabilities toward the receptor ACE2.

After careful reading there is a number of issues to be addressed. There are no critical concerns, but, in particular, it is a pity the biophysical experiments have not been done using a common buffer composition for all RBDs. This lower the significance of that comparison.

Major points:

1. An important issue not sufficiently emphasized is the fact that, very likely related to the deficient glycosylation pattern and folding when expressed in E. coli, the RBD shows very little solubility and it is expressed in inclusion bodies. This also has impact on the functionality of the refolded RBD after purification. The Table in Figure 6 (which should be a Table, not a Figure) could be completed by adding this information (solubility, or soluble expression, or need for refolding…).

2. “FACS” must be defined.

3. Any buffer with high concentration of NaCl is not recommended for circular dichroism studies, because chloride ions will preclude getting good measurements below 200 nm in the far-UV range. I understand that physiological ionic strength is usually obtained with NaCl, but, as shown in Figure 2b, experimental data are already of low quality below 210 nm. Did the authors employ lower NaCl concentration? Or other non-chloride salts?

4. Another issue is related with using different experimental conditions (buffer composition) for RBD obtained from different expression systems (PBS, pH 7.4 for RBD obtained from eukaryotic cells, but Tris, pH 8, TCEP, glycerol, for RBD obtained from E. coli). This might preclude making direct comparison (e.g., comparing far-UV CD spectra). Which are the reasons for not using similar experimental conditions.

5. It is mentioned that ΔCp is constant for a given protein. I understand the authors meant to say that ΔCp is not temperature dependent for a given protein, which is not true, but it is usually assumed for practical purposes. In addition, ΔCp may depend on other factors (e.g., pH). Then, I would suggest deleting “and it is constant for a given protein”).

6. Following the previous point, the authors say that the ΔCp value for RBD was estimated using the reported value for α-chymotrypsin. I suggest the authors report the numerical value assumed for the ΔCp of RBD.

7. Not only the glycosylation pattern, as reflected in different electrophoretic migration in Figure 1c and 1d, seem to be different when comparing RBD obtained from E. coli and eukaryotic cells, but a significant difference is also observed when comparing RBD obtained from the two types of eukaryotic cells (Figure 1d). Please, comment also on that.

8. Following the previous point, it is interesting that, despite the many differences observed between RBD obtained from different expression systems (folding, solubility, electrophoretic migration…), the unfolding temperature Tm is quite similar, pointing to a similar structural stability. Could the authors comment on this?

9. And following the previous point: very often, thermal stability studies focus only on Tm values, but the Tm provides just a minor part of the information, and very often Tm is a misleading parameter to judge and quantify stability. From the plots shown in Figure 2d it would be very easy to estimate the unfolding enthalpy, ΔHm, which together with Tm would provide an estimation of the unfolding Gibbs energy, ΔGm. Even neglecting the ΔCp-dependent term, an approximated value for ΔGm can be obtained. The stabilization Gibbs energy (or unfolding Gibbs energy) is a better index/parameter for comparing stabilities in proteins (two proteins may have similar Tm’s but different stabilization energies at 2981.5 K). ΔHm and ΔGm could reveal important differences between the RBD obtained from different expression systems.

10. And following the previous point: once the far-UV CD spectra are obtained, it would be very easy to obtain the bear-UV CD spectra. Differences in the near-UV CD spectra would point to differences in the folding (tertiary structure, disulfide bridge pattern and topology) of the different RBD obtained from different expression systems.

11. Following the previous point, in Figure 2b it can be observed a very different spectra for E. coli-derived RBD compared to the other two RBDs. Would that be a consequence of using different experimental conditions (e.g., the reducing agent TCEP). For example, the hump observed at 230 nm is very often reflection of disulfide bridges or tryptophan residues; because all RBDs studied here have the same sequence, this hump should be related to disulfide bridges. Then, does RBD from E. coli not show that feature because it did not form initially the disulfide bridges (when expressed, when refolded…), or because of TCEP in the buffer?

12. It is mentioned that the thermal denaturation of RBD obtained from E. cli showed a biphasic behavior, with an early unfolding event at 305 K. This cannot be observed in Figure 2d. And, again, would that be a consequence of the different experimental conditions?

13. The BLI plots reflect some unusual features not commonly observed (e.g., Figure 5). The signal rises abruptly, but then it enters into a moderate rising phase. As a result, it looks like two phases are occurring (i.e., two slopes in the plots). Could the authors comment on this behavior?

14. Finally, the authors summarize the results stating that, overall, the properties of E. coli-derived RBD are much worse (yield, glycosylation pattern, folding, activity…) compared to RBD obtained from eukaryotic cells (at this point, some of the differences may be due to different experimental conditions during the assays…). But, anyway, the only advantages of using E. coli seem to be the cost and the time required for getting the protein. And this is a marginal advantage (and I would say, a negligible advantage) if the final product is deficient. For example, even in the worst case, a 5-fold larger cost of producing RBD using HEK-293 cells becomes unimportant if the yield is much larger and the protein product is considerably better. Thus, using insect cells may represent the optimal alternative to E. coli or HEK-293, when considering all factors.

Round 2

Reviewer 1 Report

The authors have satisfactorily addressed the points raised by this reviewer. Two more comments: the claim that 150mM sodium chloride is a moderate concentration is relative. For a CD spectroscopist such concentration can be considered very high. I suggest rewording this as follows: " due to the presence of 150mM sodium chloride concentration". 

Second comment: NaCl can be substituted by NaF prior to CD analysis (by dialysis for example). The absorbance of fluoride ions in the far-UV region is much lower than that of chloride ions and enables data collection below the 200 nm region with a good signal/noise ratio. 

In summary, I recommend publication of the revised manuscript. 

Author Response

We thank the Reviewer for having contributed to an ameliorated version of our manuscript.

First comment:

We thank the reviewer for the useful comment. We have now corrected the sentence and replaced following the reviewer’s input.

Second comment:

We thank the revievewer for the technical suggestion. We will considering using the suggested experimental conditions for acquiring further CD data in a future work.

Reviewer 2 Report

The authors made great effort when revising the manuscript.

Author Response

We thank the Reviewer for having contributed to an ameliorated version of our manuscript.